# Chiral, air stable, and reliable Pd(0) precatalysts applicable to asymmetric allylic alkylation chemistry

Jingjun Huang[1,5], Thomas Keenan[2,5], François Richard [ID][2], Jingru Lu[1], Sarah E. Jenny[3], Alexandre Jean[4], Stellios Arseniyadis [ID][2] ✉ & David C. Leitch [ID][1] ✉

Stereoselective carbon-carbon bond formation via palladium-catalyzed asymmetric allylic alkylation is a crucial strategy to access chiral natural products and active pharmaceutical ingredients. However, catalysts based on the privileged Trost and Pfaltz-Helmchen-Williams PHOX ligands often require high loadings, specific preactivation protocols, and excess chiral ligand. This makes these reactions uneconomical, often unreproducible, and thus unsustainable. Here we report several chiral single-component Pd(0) precatalysts that are active and practically-applicable in a variety of asymmetric allylic alkylation reactions. Despite the decades-long history and widespread use of Trost-type ligands, the precatalysts in this work are the only reported examples of stable, isolable Pd(0) complexes with these ligands. Evaluating these precatalysts across nine asymmetric allylic alkylation reactions reveals high reactivity and selectivity at low Pd loading. Importantly, we also report an unprecedented Pd-catalyzed enantioselective allylation of a hydantoin, achieved on gram scale in high yield and enantioselectivity with only 0.2 mol% catalyst.

Homogeneous catalysis by transition metal complexes is one of the most powerful technologies in synthetic chemistry. Many of the most efficient and selective methods for the construction of organic molecules and materials are based on metal-catalyzed reactions, with applications encompassing bulk chemicals production, natural product synthesis, pharmaceutical manufacturing, and materials preparation. In synthetic organic chemistry, carbon-element bond forming reactions through organopalladium catalysis is without contest the most widely used strategy[1–6]. Key examples include the well-established Stille[7], Suzuki-Miyaura[8,9], Sonagashira[10], Negishi[11], Mizoroki-Heck[12,13], and Buchwald-Hartwig reactions[14], as well as the Tsuji-Trost reaction[2,15–19]. This latter example is well represented in natural product synthesis[20–24], where it enables stereoselective

$Csp^3$–$Csp^3$ and $Csp^3$–$Nsp^3$ couplings through asymmetric allylation of carbon and nitrogen nucleophiles.

A key aspect of all catalytic reactions is the generation of an active catalyst from stable precursors. In many synthetic applications involving homogeneous organometallic catalysis, this is achieved through in situ combinations of supporting ligands and a metal source that are expected to assemble into an active form. While operationally convenient, this approach often leads to inefficient catalyst generation, negatively impacting activity, reproducibility, and/or selectivity. An alternative strategy is to employ single-component precatalysts. The latter would already contain the required supporting ligands, along with carefully chosen reactive sites that lead to rapid and complete activation. Key examples include group 4 metallocene and

[1]University of Victoria, Department of Chemistry, 3800 Finnerty Road, Victoria, BC V8P 5C2, Canada. [2]Queen Mary University of London, Department of Chemistry, Mile End Road, London E1 4NS, UK. [3]Temple University, Department of Chemistry, 1901 N. Broad St, Philadelphia, PA 19122, USA. [4]Industrial Research Centre, Oril Industrie, 13 rue Desgenétais, 76210 Bolbec, France. [5]These authors contributed equally: Jingjun Huang, Thomas Keenan. ✉e-mail: s.arseniyadis@qmul.ac.uk; dcleitch@uvic.ca

post-metallocene systems for alkene polymerization[25–28]; Mo- and Ru-based complexes for olefin metathesis[29–32]; Ru-, Rh-, and Ir-based hydrogenation catalysts[33–35]; and Pd-carbene or phosphine complexes for cross-coupling reactions (Fig. 1A)[36–41].

Despite the success and widespread application of Pd-catalyzed asymmetric allylic alkylation, single-component precatalysts are rarely used for this chemistry[2,17–19,42–46]. Instead, these reactions rely on in situ catalyst generation, often using prolonged pretreatment of an achiral Pd source (e.g. Pd$_2$(dba)$_3$ or [Pd(allyl)Cl]$_2$) with excess chiral ligand to ensure complete metalation. In some cases, this also requires heating for extended periods of time, rendering the process significantly more prone to reproducibility issues. Furthermore, the use of Pd$_2$(dba)$_3$ is itself a reproducibility concern, given the known issues of quality and stability associated with this compound, with (multiple) recrystallization required to ensure high purity[47,48]. Finally, the stabilizing ligands (e.g. dba) must be removed from the desired product during purification, which can be non-trivial.

While there are examples of L*-Pd(0)-alkene complexes with phosphinooxazoline (PHOX) ligands[45,49–55], along with a few other chiral ligand classes[56–60], these are not used as precatalysts for asymmetric allylic alkylation (with one specific exception[45]). In addition, accessing *any* isolable Pd complexes of the chiral diamide bisphosphine

ligand platform – the eponymous Trost ligands – has proven particularly challenging (Fig. 1B). Early work from Trost, Breit, and Organ established that mixing (*S,S*)-$^{Ph}$DACH [**L1**, (*S,S*)−1,2-diaminocyclohexane-*N,N'*-bis(2-diphenylphosphinobenzoyl)] with Pd$_2$(dba)$_3$ generates a species consistent with [**L1**]Pd(dba), with a κ$^2$-(P,P) coordination mode; however, this species was not isolated or further characterized[61]. Furthermore, this species rapidly oxidizes when exposed to air, forming catalytically inactive bis(amidate) [PNNP]Pd$^{II}$. Further studies by Amatore, Jutand, and co-workers revealed this deactivation can happen spontaneously *even in the absence of external oxidant*[62]. Thus, Pd(0) complexes of the Trost ligands were considered unstable and therefore unsuitable as precatalysts, as the [PNNP]Pd$^{II}$ species is known to be catalytically inactive[63].

As part of extensive studies on the solution behavior of catalytically relevant Pd species in asymmetric allylation reactions, Lloyd-Jones and co-workers reported a dipalladium(II) complex with **L1**, where each Pd center is coordinated by one phosphine and one carbonyl oxygen ([**L1**]Pd$_2$(allyl)$_2$(OTf)$_2$, Fig. 1B)[42]. While this compound is catalytically active, it gives essentially racemic product during an attempted kinetic resolution of (±)-cyclopent-2-en-1-yl pivalate with NaCH(CO$_2$Me)$_2$, likely due to the κ$^2$-(P,O) rather than κ$^2$-(P,P) coordination mode. A monopalladium compound of the empirical formula

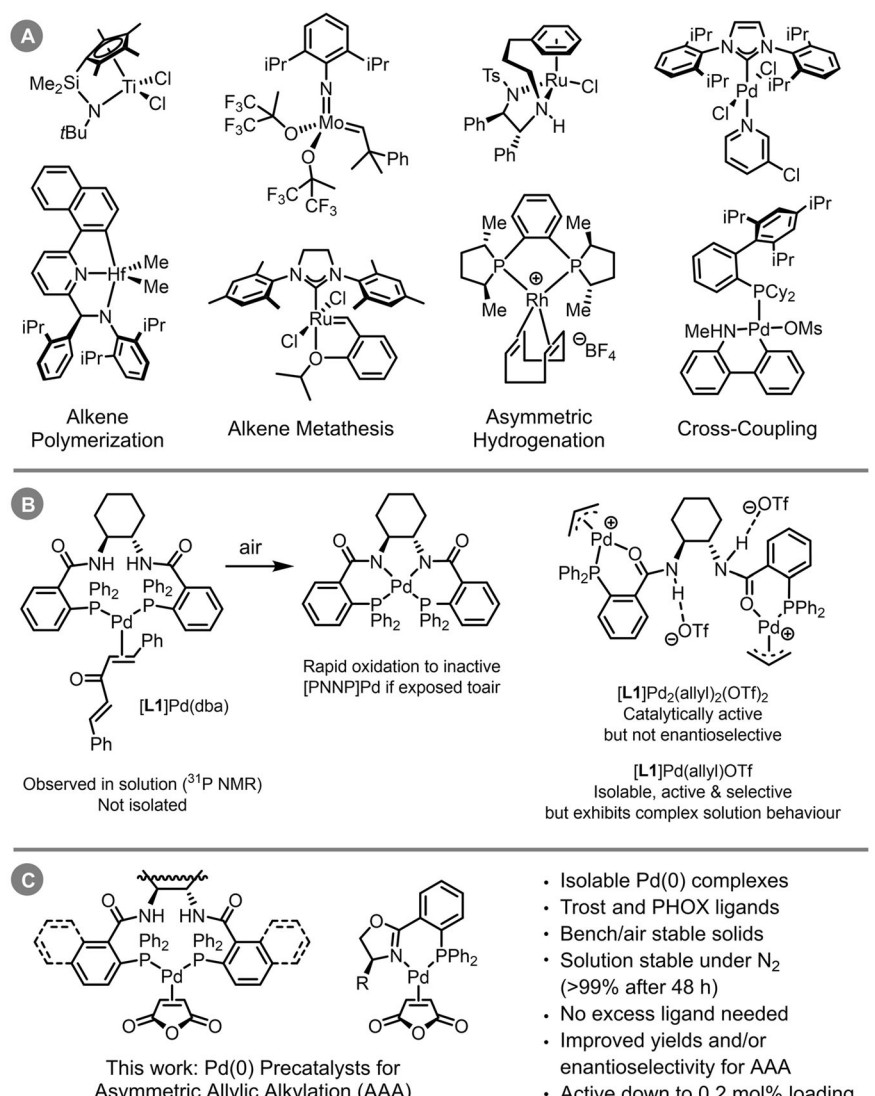

**Fig. 1 | Homogeneous catalysts in organic chemistry. A** Exemplar organometallic precatalysts for specific transformations from across the periodic table. **B** Challenges with accessing precatalysts containing Trost-type ligands, including

rapid oxidation of Pd(0) species derived from Pd$_2$(dba)$_3$, and complex solution behavior of Pd(II) allyl species. **C** This work: single-component Pd(0) precatalysts for asymmetric allylic alkylation.

[**L1**]Pd(allyl)(OTf) can be isolated when using a 1:1 stoichiometry of **L1** to [Pd(allyl)(MeCN)₂][OTf]. While this compound is both active and enantioselective, it is not well-defined, exhibiting concentration-dependent solution behavior. Multiple Pd-containing species can be present, including [**L1**]Pd₂(allyl)₂(OTf)₂ and higher oligomers[42,44]. To the best of our knowledge, [**L1**]Pd(allyl)(OTf) is the only example of an isolable single-component precatalyst based on the Trost ligand platform, though the complexities of its synthesis and characterization seem to have precluded its wider use.

Here we report a class of stable and isolable chiral Pd precatalysts for asymmetric allylic alkylations (Fig. 1C). These Pd(0) complexes, which are based on either Trost-type or PHOX-type ligands, are easily prepared, well defined, monomeric, and easily handled without the need for an inert atmosphere or glovebox. Importantly, the oxidative degradation exhibited by [**L1**]Pd(dba) is not a significant issue with these precatalysts. In addition, they are highly effective as single-component precatalysts for a multitude of asymmetric allylic alkylation reactions, operating with improved yield, selectivity, catalyst loading, and/or operational simplicity relative to established systems. The advantages of these compounds as precatalysts are exemplified not only in known allylation chemistry, but also by their ability to catalyze the unprecedented enantioselective allylation of a

hydantoin-derived nucleophile, exhibiting high yield and stereoselectivity with only 0.2 mol% catalyst on gram scale (TON = 465).

## Results

### Precatalyst synthesis and characterization

Given the aforementioned challenges with accessing discrete, stable Pd precatalysts with Trost-type ligands, we targeted these Pd(0) species first. Using ᴰᴹᴾDAB-Pd-MAH – a Pd(0) source we recently reported[64] – we examined ligand substitution reactions with a set of six chiral ligands (**L1**–**L6**, Fig. 2). In each case, reaction monitoring by ¹H and ³¹P NMR spectroscopy (with internal standard) revealed complete consumption of ᴰᴹᴾDAB-Pd-MAH and quantitative formation of L*-Pd-MAH (**1**–**6**) in <20 min at room temperature; this rapid metalation is also evident by a near-immediate color change from red/purple to yellow.

On preparative-scale, **1-6** can be easily isolated and purified by simple precipitation to remove the soluble ᴰᴹᴾDAB, giving the chiral Pd(0) complexes in 66–90% isolated yield after purification. We have prepared up to 340 mg of **1** in a single batch, demonstrating a path to scalable synthesis. These complexes are derived from the most common chiral ligands used in asymmetric allylation: four Trost ligands (**L1**–**L4**), and two PHOX ligands (**L5** and **L6**). In addition to full

**Fig. 2 | Synthesis of chiral Pd(0) precatalysts** 1–6 **via ligand substitution of** ᴰᴹᴾ**DAB-Pd-MAH. Solid-state molecular structures (X-ray diffraction, 50% probability ellipsoids, Ph groups on phosphorus shown in wireframe for clarity) shown for compounds 2, 4, and 6; crystal/diffraction data and metrical** **parameters are in Supplementary Information.** Intramolecular H-bond in **2** indicated with green dashed line, NH---O distance = 2.22 Å. Intermolecular H-bond in **4** between amide N–H and THF molecule indicated with green dashed line, NH---O_THF distance = 2.23 Å.

characterization by multinuclear NMR spectroscopy (*vide infra* and SI) and high-resolution mass spectrometry, we have obtained solid-state molecular structures by X-ray crystallography for three complexes (**2, 4**, and **6**). Surprisingly, complexes **1-4** represent the only characterized examples of isolable Pd complexes containing Trost-type ligands bearing the desired κ²-(P,P) binding mode. This is especially noteworthy given the long and extensive history of these ligands in Pd catalysis.

The solid-state molecular structures of **2** and **4** reveal not only the desired κ²-(P,P) coordination mode, but also two distinct conformations. In complex **2**, the MAH binds with the carbonyl groups pointing toward the chiral tether of the <sup>NAP</sup>DACH ligand (*endo* conformation). This enables an intramolecular hydrogen bond between an amide N−H and a carbonyl oxygen from MAH. Importantly, this type of hydrogen bond between ligand and substrate is proposed as a critical intramolecular interaction in the mechanism for stereoinduction in many asymmetric allylic alkylation reactions[65]. Here, we directly observe this interaction in both the solid state and in solution (*vide infra*). In contrast, the <sup>Ph</sup>ANDEN complex **4** has the MAH oxygens pointing away from the chiral tether (*exo* conformation). Notably, there is an intermolecular hydrogen bond observed between one of the amides N−H and a co-crystallized THF molecule. These alternative geometries are due to the flexibility of the large macrocyclic chelate in these Trost-type complexes.

Solution-phase characterization of complexes **1–6** by multinuclear and multidimensional NMR spectroscopy reveals the presence of two distinct species in every case. For both PHOX-based complexes **5** and **6**, these two species are present in an approximately 1:1 ratio, as indicated by ¹H and ³¹P NMR spectroscopy in either $d_2$-DCM or $d_8$-THF. We attribute this solution behavior to the presence of both *endo* and *exo* conformers, with no clear energetic preference for either[45,53]. This conformer assignment is supported by 2D NMR spectroscopy characterization data, including 2D NOESY (see SI). For complex **1**, ³¹P NMR spectra obtained in either $d_2$-DCM or $d_8$-THF also contain two sets of signals, each of which is a matching pair of doublets that is characteristic of bidentate κ²-(P,P) coordination to Pd (Fig. 3A). In DCM, the two sets of signals have a 55:45 peak area ratio; in contrast, in THF there is a 14:1 peak area ratio between the major and minor signals. As for **5** and **6**, we attribute this to the presence of two distinct conformers, the ratio of which has a clear solvent dependence.

Based on these data and the solid-state molecular structures of **2** and **4**, we propose that the two species in solution are conformers **1-endo** and **1-exo** (Fig. 3B), with **1-exo** being the major (based on extensive NMR spectroscopy, Supplementary Figs. 1–19). To explain these observations, we propose that in a relatively non-polar solvent (*e.g.* DCM), **1-endo** and **1-exo** are similar in energy; however, with the addition of a hydrogen-bond-accepting solvent (e.g. THF), hydrogen bonding between an N−H on the ligand and the solvent will stabilize the **1-exo** conformer. Notably, complexes **2–4** exhibit similar NMR spectroscopic characteristics, consistent with *endo* and *exo* conformers (see SI for further details).

Finally, to support our conformer assignments, we calculated relative electronic energies of the **1-exo** and the **1-endo** conformers (RI-B2PLYP-D3BJ/def2-TZVP//RI-BP86-D3BJ/def2-SVP, with def2-TZVP/C and def2/J auxiliary basis sets for the RI part, respectively) using conductor-like polarizable continuum (CPCM) implicit solvation models (Fig. 3C)[66]. In gas phase calculations, both conformers exhibit an intramolecular hydrogen bond between the amides, and the **1-endo** conformer has an additional hydrogen bond between an amide N−H and the MAH (as observed in the solid-state structure of **2**). This leads to the **1-endo** conformer being 10.5 kJ mol⁻¹ more stable than **1-exo**. However, the **1-exo** and **1-endo** energies calculated in DCM solvent are very close, with **1-exo** only slightly more stable (0.5 kJ mol⁻¹ difference). To compare energies in THF, we used both the CPCM implicit model and included one explicit THF molecule

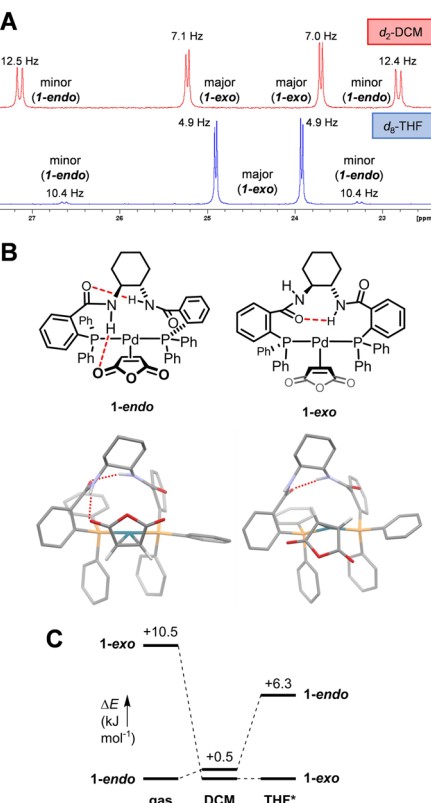

**Fig. 3 | Conformational analysis of compound 1. A** ³¹P NMR spectra of complex **1** in $d_2$-DCM and $d_8$-THF, revealing two conformers (**1-exo** and **1-endo**). **B** Calculated structures of **1-endo** and **1-exo** (gas phase geometry shown). **C** Relative electronic energies of **1-exo** and **1-endo** in the gas phase, DCM, and THF (implicit solvation models; one explicit THF molecule included in THF-solvation calculations).

hydrogen-bonding to a ligand N−H in **1-exo** (as observed in the solid-state structure of **4**). This results in **1-exo** being 6.3 kJ mol⁻¹ more stable. These calculations support not only the assignment of **1-exo** as the major conformer in THF, but also exhibit the same solvent effect trend as our spectroscopic observations.

To assess whether complexes **1–6** could be robust and user-friendly precatalysts, we examined their solution stability in THF (15–50 mg/mL) over 48 h by ³¹P NMR spectroscopy (Fig. 4). With solutions prepared under N₂, the concentrations of all six complexes remain unchanged over this period. Given the aforementioned rapid oxidation of [**L1**]Pd(dba) to the tetradentate [PNNP]Pd<sup>II</sup> species, we also assessed the stability of a THF solution of complex **1** after exposure to air. After 48 h, there is still >80% of **1** intact, with the mass balance comprised of [PNNP]Pd<sup>II</sup>. In stark contrast, a mixture of **L1** and Pd₂(dba)₃·CHCl₃ is 50% oxidized after only 30 min, and is completely converted to [PNNP]Pd<sup>II</sup> within 18 h. As a solid, complex **1** is very stable when stored under N₂ at room temperature (2 years thus far). Complex **1** is even stable for weeks as a solid under air (<6% area of new signals observed in ³¹P NMR spectra after one month). Thus, precatalysts **1–6** can be handled and used without the need for a glovebox. In fact, all catalytic evaluations were carried out by weighing the precatalysts and preparing solutions under air (*vide infra*).

## Deploying single component precatalysts in asymmetric allylic alkylations

To conduct an initial assessment of the performance of precatalysts **1–6** in diverse asymmetric allylic alkylations, we compared them against standard catalyst systems for a series of benchmark reactions. Representative comparisons at identical Pd loading are given

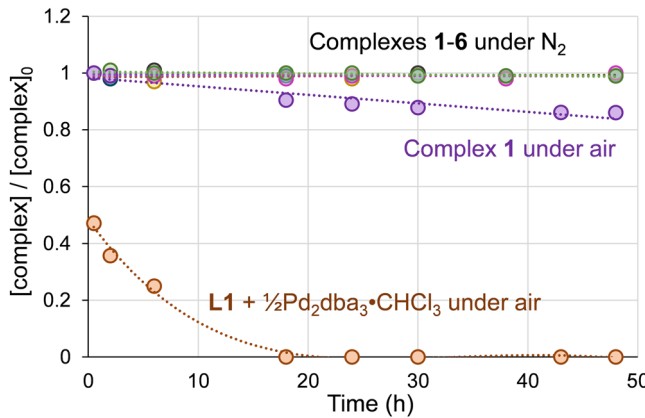

**Fig. 4 | Concentration versus time plot for complexes 1–6 in THF under N₂, showing no decomposition over at least 48 h, as well as complex 1 and L1 + Pd₂(dba)₃•CHCl₃ under air.** These latter experiments show slow decomposition of **1** (>80% intact after 48 h) and rapid decomposition of [**L1**]Pd(dba) (<50% remaining after 30 min).

in Fig. 5; additional comparator data under alternative reaction conditions is available in Supplementary Tables 5–8. We chose five criteria to evaluate each catalyst system, visualized in a quantitative radar plot analysis, including isolated yield, enantioselectivity, reaction time, the Pd to chiral ligand ratio, and the Pd / ligand premixing time. In every case, catalytic reactions were set up and executed without the use of a glovebox.

We initially evaluated the extensively studied malonate allylation with racemic cyclohex-2-en-1-yl methyl carbonate (**7**) (Fig. 5A, Supplementary Table 5)[65,67]. Under standard reaction conditions (i.e. 5 mol% Pd, a Pd/L ratio of 1:1.5, and a 30 min pretreatment of Pd precursor and ligand), the allylation proceeds efficiently and selectively using Pd₂(dba)₃•CHCl₃ to give **8** (96% yield, 95% *ee*). However, simply reducing the Pd/L ratio to 1:1 results in no observable reaction. This somewhat shocking result is not only undesirable from a cost perspective (the additional wasted chiral ligand), it indicates a potential and severe failure mode if insufficient **L1** is used. In sharp contrast, the use of ^PhDACH-Pd-MAH (**1**) as a single-component precatalyst, which has a 1:1 Pd/L ratio and does not require any pretreatment to ensure complete metalation, delivers **8** in 86% isolated yield and 96% *ee*.

A classic example in asymmetric allylation is the desymmetrisation of *meso*−2-en-1,4-diol diester **9** using the Pd/**L1** system (Fig. 5B, Supplementary Table 6). We employed the *bis*(acetate) substrate **9** to accentuate reactivity differences between the systems, as it is known to be less reactive than the more commonly used *bis*(benzoate) derivative[68,69]. At a Pd loading of 2 mol% and a Pd/**L1** ratio of 1:1, Pd₂dba₃•CHCl₃ performs poorly (15% yield, 92% *ee*) while [Pd(allyl)Cl]₂ generates **10** in high enantiopurity but only modest yield (46% yield, 99% *ee*). ^PhDACH-Pd-MAH (**1**) is superior to these known systems, exhibiting excellent enantioselectivity with improved yield, all without catalyst preactivation (68% yield, 98% *ee*).

Next, we examined the Pd-catalyzed asymmetric allylation of phthalimide (**11**) as a nitrogen-based nucleophile using racemic epoxide **12** (Fig. 5C, Supplementary Table 7); this transformation has been utilized in several total syntheses[70–72]. Using [Pd(allyl)Cl]₂ or Pd₂(dba)₃•CHCl₃ as the Pd source with ^NAPDACH (**L2**) as the ligand (0.8 mol% Pd, 1:1.5 ratio of Pd/**L2**), we obtained good yields of the homoallyl alcohol **13** (99% and 82% respectively); however, we were not able to achieve the reported level of enantioselectivity even after rigorous purification of every component of the reaction mixture (72% and 64% versus the reported 96%). The reduced enantioselectivites observed for these Pd sources, especially when using Pd₂dba₃•CHCl₃,

may be due to a lower effective catalyst concentration due to oxidative decomposition to [PNNP]Pd^II. Trost et al. reported that the enantioselectivity of this reaction is sensitive to catalyst loading, such that "lowering the catalyst further [below 0.4 mol%] significantly decreased the *ee*."[70,72] Amatore et al. reported that **L2**–Pd–dba converts to the corresponding [PNNP]Pd^II even in the absence of dioxygen. Notably, the single-component precatalyst ^NAPDACH-Pd-MAH (**2**) achieves excellent yield while attaining higher enantioselectivity than the in situ systems, without the need for excess **L2** or preactivation (99% yield, 81% *ee*). This result highlights the robustness of our Pd(0) precatalysts, which are effective even for reactions that are clearly sensitive to the specific reaction conditions.

Decarboxylative asymmetric allylic alkylation (DAAA) reactions represent a broad class of Pd-catalyzed allylation chemistry, and hence an important testing ground for our precatalysts[73,74]. We therefore examined Pd-catalyzed DAAA of allyl (2-phenyl-cyclohexyl) carbonate (**14**, Fig. 5D, Supplementary Table 8) using the (*S,S*)-^PhANDEN (**L4**) based catalyst. At 2 mol% Pd loading and 3 mol% of **L4**, Pd₂(dba)₃·CHCl₃ is able to achieve complete conversion in 30 min, giving **15** in 77% yield with good enantioselectivity (83% *ee*). However, this system fails to reach completion even after 24 h when the Pd/**L4** ratio reduced to 1:1 (39% yield). In sharp contrast, the single-component ^PhANDEN-Pd-MAH (**4**) complex achieves complete conversion and good enantioselectivity without requiring an excess of **L4** (87% yield, 81% *ee*).

To assess PHOX-type precatalysts **5** and **6**, we used the reaction of racemic allyl acetate **16** with dimethyl malonate, which is often used as a model reaction when developing Pd catalysts for asymmetric allylation chemistry. Hence, we compared [Pd(allyl)Cl]₂ and Pd₂(dba)₃·CHCl₃ with complexes **5** and **6** for the synthesis of **17** (Fig. 6, Supplementary Table 9)[51,75]. Using conditions for mild enolate generation with bis(trimethylsilyl)acetamide (BSA) and catalytic KOAc, and the established method of premixing the Pd source and PHOX ligand (1.25 equiv. per Pd) at 50 °C for 1 h to ensure complete metalation, [Pd(allyl)Cl]₂ delivered **17** in 93% yield and 95% *ee* after 1 h (2 mol% Pd). However, if the 1 h preactivation step is conducted at room temperature, [Pd(allyl)Cl]₂ suffers from diminished yield even after 24 h (67%). Interestingly, the single-component chiral precatalyst (*S*)-^iPrPHOX-Pd-MAH (**6**), which can simply be added as a solid to the reaction mixture, requires no preactivation or excess chiral ligand and provides **17** in 84% isolated yield and 96% *ee* (opposite enantiomer to that obtained with (*R*)-^iPrPHOX), though the reaction requires 24 h to reach completion. Use of the ^tBuPHOX ligand resulted in similar outcomes, where [Pd(allyl)Cl]₂ outperforms the MAH-based catalyst **5** in terms of rate, but the in situ system again requires preactivation at 50 °C.

In contrast to the weakly basic BSA/KOAc reaction conditions, use of KH to generate a harder potassium enolate results in excellent reaction rates for all three catalyst systems (Fig. 6B), with the single-component precatalysts **5** and **6** matching the reactivity of the [Pd(allyl)Cl]₂ based in situ system (1 h reaction time for complete conversion in each case). In addition, both **5** and **6** exhibit excellent enantioselectivity. The increase in reaction rate for the MAH-based systems under strongly basic conditions indicates that the catalyst activation processes are likely dependent on the reaction conditions.

## Single component precatalysts enable asymmetric allylation of butenolide nucleophiles

To explore the suitability of our precatalysts beyond well-known allylation chemistry, we turned toward the preparation of enantioenriched heterocycles relevant to the synthesis of natural products and commodity chemicals. Specifically, the enantioselective functionalization of prochiral heterocycles is a vast field with scores of possible methodologies[76–79]. In Pd-catalyzed asymmetric allylation, there are three predominant methods: a) the direct allylation of

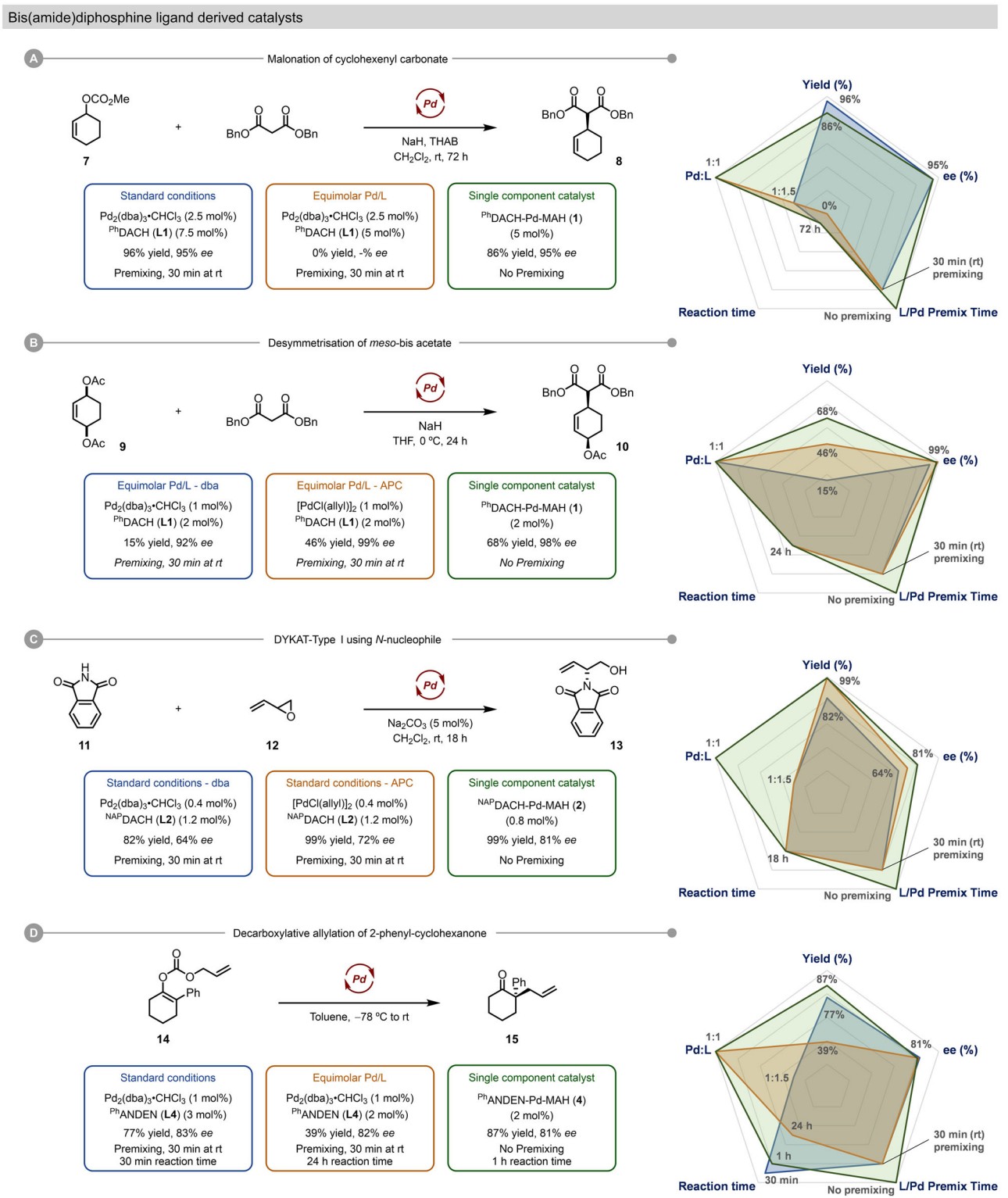

**Fig. 5 | Comparison of established catalyst systems with single-component precatalysts for benchmark AAA reactions. A** Malonation of cyclohexenyl carbonate. **B** Desymmetrization of *meso* bis(acetate). **C** Dynamic kinetic asymmetric transformation (DYKAT) of racemic epoxide **12** with phthalimide (**11**). **D** Intramolecular decarboxylative asymmetric allylic alkylation (DAAA).

a prochiral heterocycle (e.g. **18**, Fig. 7A); b) the decarboxylative allylation (e.g. **20**, Fig. 7B); or c) the allylation of the corresponding enol silane (e.g. **21**, Fig. 7C). We explored the formation of chiral butenolide **19** by each of these methods, comparing in situ catalyst formation using Pd$_2$(dba)$_3$·CHCl$_3$ to the use of single-component chiral precatalyst **1**[80,81].

The direct allylation of **18** and the decarboxylative allylation of **20** did proceed using high Pd$_2$(dba)$_3$·CHCl$_3$ loading (Supplementary Tables 10 and 11); however, both approaches were ineffective when lowering the Pd loading from 10 to 2 mol%, resulting in little to no reactivity even after extended reaction times (Fig. 7A, B). In sharp contrast, $^{Ph}$DACH-Pd-MAH (**1**) operates at 2 mol% loading to give **21** in

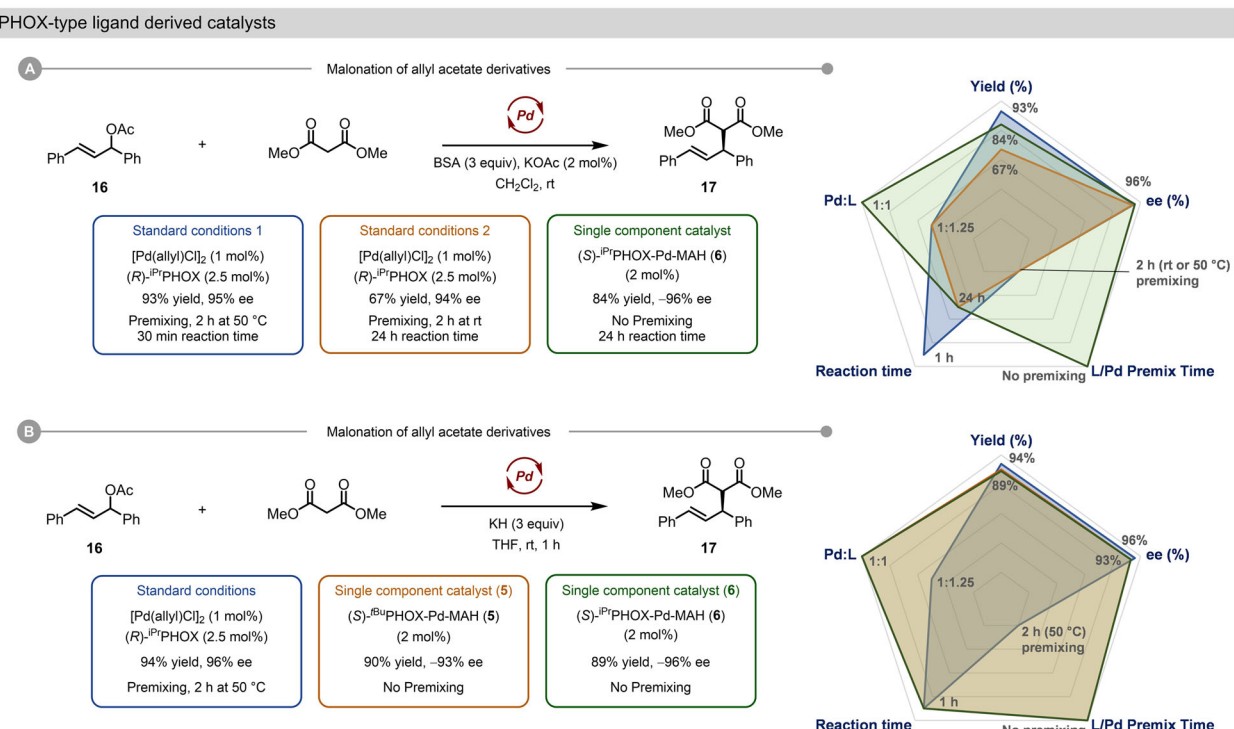

**Fig. 6 | Comparison of established PHOX-type catalyst systems with single component precatalysts for benchmark AAA reactions. A** Malonation of allyl acetate derivatives using weakly basic conditions for soft enolization. **B** Malonation of allyl acetate derivatives using strongly basic conditions to generate a potassium enolate nucleophile.

good yield (65-75%) and enantiopurity (82% *ee*) under both conditions. While the direct allylation reaction time is long (72 h), **1** is able to catalyze complete decarboxylative allylation to form **19** within 4 h.

In the third approach, involving allylation of the enol silane **21**, Pd₂(dba)₃·CHCl₃ is able to operate at the lower 2 mol% Pd loading, giving full conversion and good yields after only 15 min; however, this system failed when the Pd/L ratio was reduced to 1:1 (Fig. 7C). This was circumvented by using 2 mol% of ᴾʰDACH-Pd-MAH (**1**) under otherwise identical conditions, giving comparable yield and selectivity without the need for excess chiral ligand. Overall, the ability to reliably reduce both Pd and L loading in these transformations, and avoid pretreatment steps, are significant advantages in cost, robustness, and operational simplicity.

## Precatalyst-enabled discovery and optimization of the enantioselective allylation of hydantoins

The rapid in situ metalation of Trost-type and PHOX-type ligands to the ᴰᴹᴾDAB-Pd-MAH precursor, and the stability and ease-of-use for the single-component precatalysts **1**–**6**, render these systems particularly suited to reaction discovery and optimization via parallel experimentation techniques. Indeed, the remarkable air stability of the L*-Pd-MAH complexes both in the solid state and in solution makes setting up array-based experiments operationally simple without the need for a glovebox. To demonstrate this feature of our precatalysts, we employed them in the rapid optimization of a Pd-catalyzed asymmetric allylation of hydantoin **22** (Fig. 8). Hydantoin heterocycles are common motifs in FDA approved drugs; however, there are a striking lack of methods for their asymmetric functionalisation[82–85]. To the best of our knowledge, there are no prior reports of enantioselective hydantoin allylation using Pd catalysis. If realized, this would be a powerful method to install a stereogenic tetrasubstituted carbon onto a protected amino acid motif.

Our initial set of experiments (Round 1) used ᴰᴹᴾDAB-Pd-MAH (2 mol%) as the source of Pd(0) to screen chiral ligands for the

proposed reaction using in situ catalyst formation (Pd/L, 1:1.5) Notably, only (*S,S*)-ᴾʰDACH gave a good enantioselectivity (70% *ee*), with all other ligands providing poor selectivity. Round 2 screening utilized chiral complex (*S,S*)-ᴾʰDACH-Pd-MAH (**1**, 2 mol%) in a solvent/base array to rapidly optimize the reaction conditions at room temperature. While the use of the single-component chiral precatalyst did increase the yield to 90% using NaHMDS/THF, all other solvent/base combinations either failed to give product in >40% yield and/or failed to improve the enantioselectivity. To further optimize the enantioselectivity and catalytic efficiency of this reaction, we reduced the catalyst loading, lowered the reaction temperature, and increased the reactant concentration (Round 3). Consistent with previous reports showing that higher catalyst concentrations can decrease stereoselectivity, we observe improved % *ee* when lowering the Pd loading from 1 mol% (76% *ee*) to 0.2 mol% (86% *ee*)[42–44,76]. We did not observe any improvement of the enantioselectivity when carrying out the reaction at −40 °C, nor when lowering further the Pd loading to 0.1 mol%. Finally, increasing the overall reaction concentration to 0.2 M and maintaining a Pd catalyst loading of 0.2 mol% led to 90% yield and 88% *ee*. Translating these reaction conditions to a gram scale (4 mmol) preparation gave 93% isolated yield of **23** with 88% *ee*, corresponding to a catalyst TON of 465. Notably, only 7.2 mg of precatalyst **1** was required to produce 1.34 g of the desired product **23**. Indeed, such low chiral catalyst loading is rare for Pd-catalyzed asymmetric allylation chemistry, and clearly demonstrates the power of using a single-component catalyst system for both screening and preparative-scale synthesis.

## Discussion

We have developed a set of bench stable chiral Pd(0) precatalysts based on the two key ligand classes used in Pd-catalyzed asymmetric allylation chemistry. By exploiting the rapid ligand substitution chemistry exhibited by the ᴰᴹᴾDAB-Pd-MAH precursor, Pd complexes with Trost-type and PHOX-type ligands can either be generated in situ, or isolated as single-component precatalysts. These complexes have

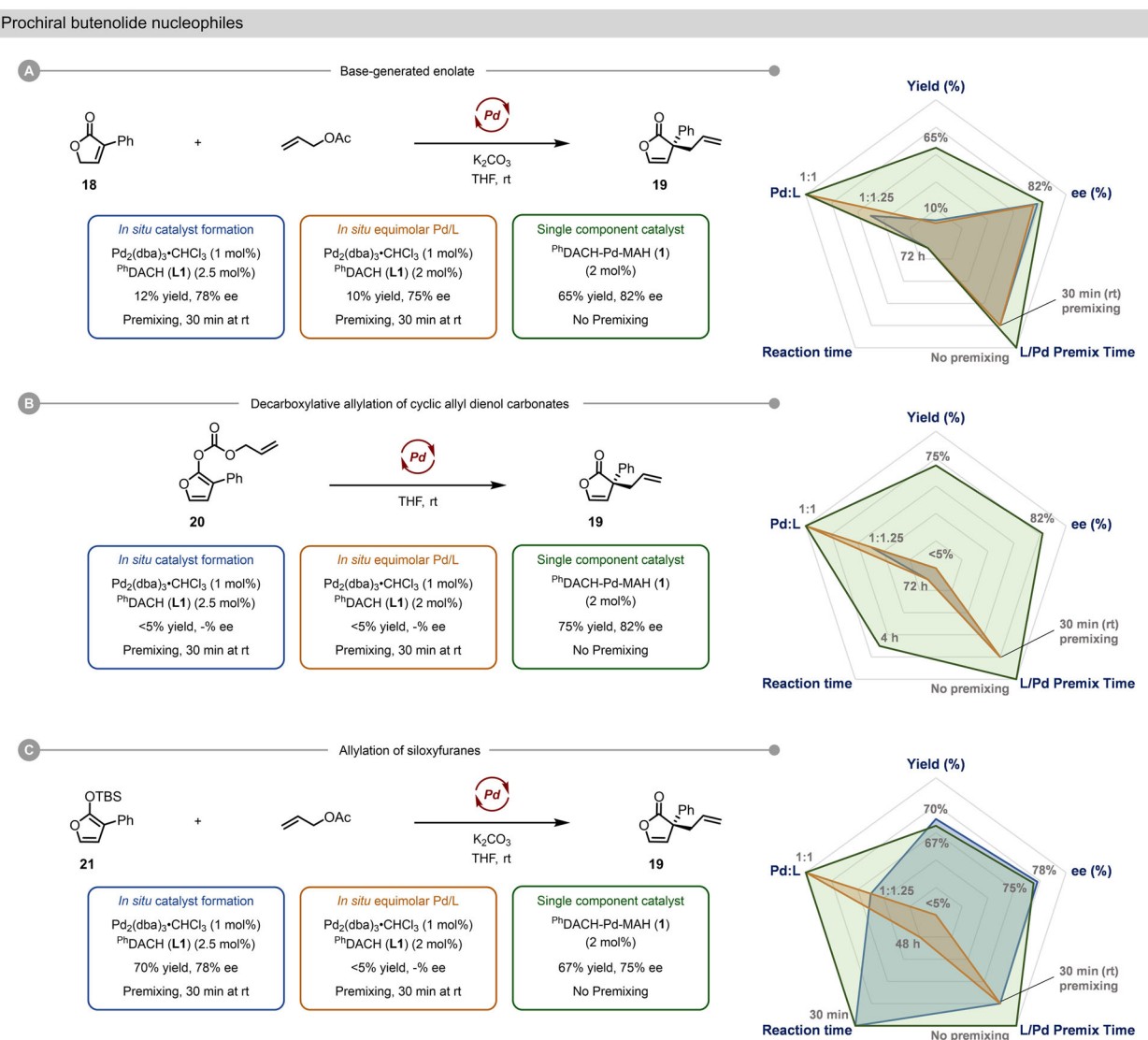

**Fig. 7 | Comparison of precatalyst 1 to in situ systems for challenging asymmetric allylic alkylation reactions in the synthesis of chiral butenolide 19. A** Direct allylation of **18**. **B** Decarboxylative allylation of **20**. **C** Allylation of siloxyfuran **21**.

been fully characterized, including X-ray diffraction studies on three examples, enabling interrogation of their conformations in solution and the solid-state. This includes direct observation of an intramolecular hydrogen bond between an amide N–H and a C=O of the coordinated maleic anhydride, which models a key interaction in the proposed mechanism for stereoinduction with the Trost ligand class. Importantly for their application in catalysis, the Trost-type complexes **1–4** do not suffer from rapid decomposition via oxidation, and can be handled as solids or even in solution under air.

To demonstrate the potential of these precatalysts, we have applied them to nine distinct Pd-catalyzed asymmetric allylation reactions. In all cases, these precatalysts compared favorably with traditional catalytic systems, maintaining enantioselectivity while enabling lower catalyst loadings. In addition, the single-component nature of **1–6** gives a perfectly controlled Pd/L stoichiometry, and circumvents the need to ensure ligand metalation is complete prior to substrate addition.

To exemplify the power of these precatalysts, we applied them to realize the heretofore unprecedented enantioselective allylation of a hydantoin derivative at only 0.2 mol% catalyst loading. The discovery and optimization of this reaction was enabled by microscale array-based experiments, to which our precatalysts are ideally suited.

Overall, the selectivity, activity, stability, and practicality of these single-component precatalysts make them powerful and attractive alternatives to established in situ catalyst generation procedures. Complexes **1–6** and analogs thereof should therefore be evaluated as standard systems for Pd-catalyzed asymmetric allylations in academic and industrial applications alike.

## Methods

### General procedure for the synthesis of the single component chiral catalysts

These reactions were set up in a glovebox under an inert nitrogen atmosphere due to the potential oxygen-sensitivity of the phosphine ligands. A representative procedure for the synthesis of **1** is given here, with specific procedures for all precatalysts given in Supplementary Information. A 4 dram (~15 mL) vial was charged with DMPDAB-Pd-MAH (200.1 mg, 0.43 mmol), **L1** (294.8 mg, 0.43 mmol, 1.0 equiv), anhydrous inhibitor-free THF (7 mL) and a cross-shaped magnetic stirbar. The reaction mixture was stirred at room temperature for 2 h. During this time, the solution changes color from an initial dark red/purple to yellow. The solvent was then removed in vacuo to give a yellow solid residue. This residue was mixed thoroughly with hexanes/diethyl ether (1:1), followed by decantation of the liquid phase (with or without

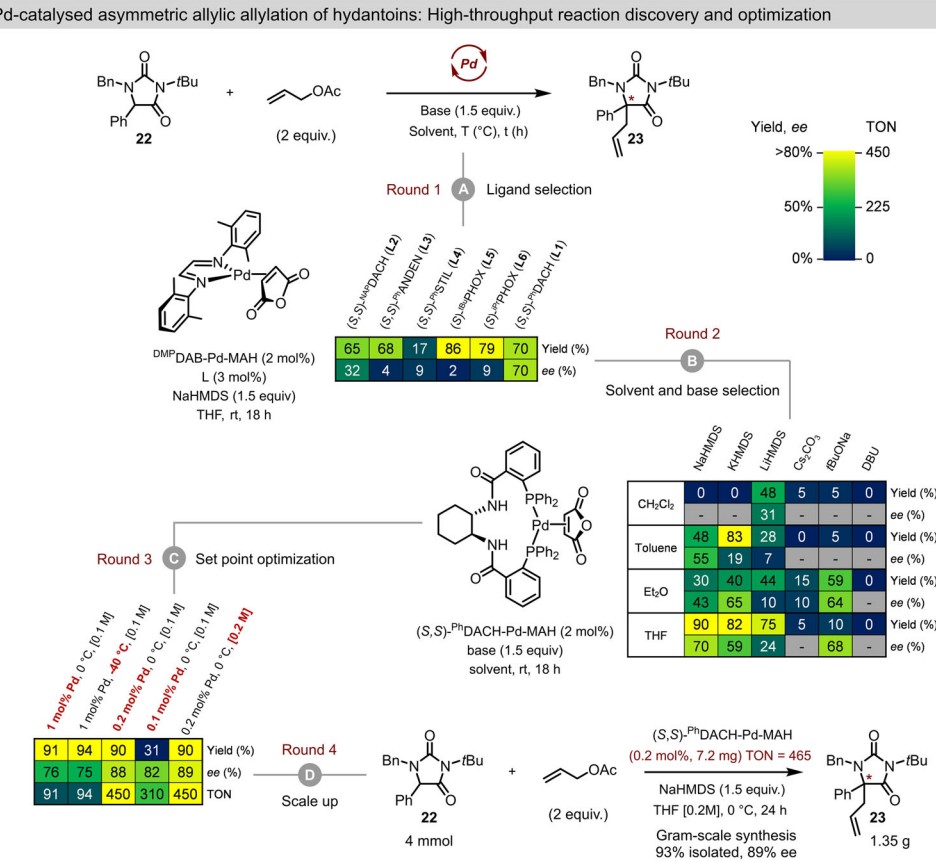

**Fig. 8 | Application of MAH-based Pd precatalysts in reaction discovery and optimization for the asymmetric allylation of hydantoin derivative 22.** At low Pd loading. **A** Ligand selection achieved using in situ catalyst formation with ${}^{DMP}$DAB–Pd–MAH. **B** 24 reaction solvent/base array using precatalyst **1**. **C** Optimization of continuous variable setpoints to maximize yield, enantioselectivity, and catalyst TON. **D** Gram-scale synthesis of **23** using optimized conditions.

centrifugation as required). This trituration/decantation process was repeated 5 more times to remove the ${}^{DMP}$DAB byproduct as well as any excess phosphine ligand. The solid was then dried *in vacuo* to give the corresponding single component chiral catalyst **1** as a pale yellow solid (339.0 mg, 89% yield).

### Procedure for the gram-scale synthesis of 23

This reaction was set up without use of a glovebox using standard air-free techniques involving nitrogen streams and balloons. An oven-dried 50 mL round-bottom flask was charged with **22** (1.29 g, 4.00 mmol) and a teflon-coated stirbar. The flask was sealed with a rubber septum and placed under nitrogen atmosphere. THF (14 mL) was added, and the contents cooled to 0 °C. NaHMDS (1 M in THF, 6.00 mL, 6.00 mmol, 1.5 equiv) was added via syringe with stirring, and the mixture was stirred for 1 h at 0 °C. ${}^{Ph}$DACH-Pd-MAH (**1**) was then added as a solid (7.2 mg, 0.0080 mmol, 0.2 mol%), followed by allyl acetate via syringe (0.860 mL, 8.00 mmol, 2 equiv). The mixture was stirred at 0 °C for 24 h, after which time TLC analysis indicated complete conversion. The reaction was quenched with the addition of H₂O (50 mL), followed by extraction with CH₂Cl₂ (3 ×50 mL). The combined organic layers were then washed with 10% wt/wt aqueous citric acid (2 ×50 mL) and brine (50 mL). The organic phase was dried over MgSO₄, filtered, and the solvent removed under reduced pressure. The crude product was purified by column chromatography using silica gel and hexane/EtOAc eluent (100:0 to 80:20 v/v) to yield **23** (1.35 g, 89% *ee*). Enantiopurity was measured by HPLC using a hexane/isopropanol mobile phase (isocratic 9:1 v/v, 1 mL/min) and a Daicel CHIRALPAK™ IC column (250 ×4.6 mm; 5 µm; 35 °C).

Specific procedures for all catalytic reactions are given in the Supplementary Information.

## Data availability
Source data are present. All processed spectroscopic and chromatographic data from the present study is contained in the Supplementary Information. The atomic coordinates for structures generated by DFT calculations of **1**-*exo* and **1**-*endo* are provided in xlsx format as a Source Data File. The X-ray crystallographic coordinates for structures reported in this study have been deposited at the Cambridge Crystallographic Data Centre (CCDC), under deposition numbers 2258957-2258960. These data can be obtained free of charge from The Cambridge Crystallographic Data Centre via www.ccdc.cam.ac.uk/data_request/cif. Source data are provided with this paper.

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

## Acknowledgements

J.H., T.K. and D.C.L. acknowledge with respect the Lekwungen peoples, on whose traditional territory the University of Victoria (UVic) stands, and the Songhees, Esquimalt, and WSÁNEĆ peoples whose historical relationships with the land continue today. They also thank Prof. Irina Paci at UVic for fruitful discussions regarding computational studies. D.C.L. thanks UVic (start-up funding), NSERC (RGPIN-2019-04985 and I2IPJ 561560 – 21), and CFI / BCKDF (JELF 38750) for general operating and equipment funds. S.A. and T.K. thank Dr. Rodolphe Tamion and Dr. Jean Fournier for fruitful discussions. S.A. thanks Dr. Lucile Vaysse-Ludot from Oril Industrie affiliated to Les Laboratoires Servier and Queen Mary University of London for financial support. S.E.J. thanks Dr. William A. Sabbers, Dr. Taylor M. Keller, and Prof. Michael J. Zdilla for fruitful discussions regarding X-ray crystallography.

## Author contributions

J.H., T.K., D.C.L. and S.A. conceived and designed the study. J.H., T.K. and F.R. performed the synthetic experiments and analyzed the data for all compounds. J.L. performed computational analysis. S.E.J. conducted single crystal X-ray diffraction experiments and analysis. A.J., S.A. and D.C.L. supervised the research. J.H., T.K., F.R., S.A. and D.C.L. co-wrote the paper with input from all authors.

## Competing interests

The authors declare the following competing financial interest(s): PCT international patent applications have been filed based partly on this work (WO2022153180A1, WO2023156972A1). $^{DMP}$DAB-Pd-MAH is commercially available from MilliporeSigma (product number 922889).
